# SeaLion: Semantic Part-Aware Latent Point Diffusion Models for 3D Generation

## Abstract

Denoising diffusion probabilistic models have achieved significant success in point cloud generation, enabling numerous downstream applications, such as generative data augmentation and 3D model editing. However, little attention has been given to generating point clouds with point-wise segmentation labels, as well as to developing evaluation metrics for this task. Therefore, in this paper, we present *SeaLion*, a novel diffusion model designed to generate high-quality and diverse point cloud with fine-grained segmentation labels. Specifically, we introduce the *semantic part-aware latent point diffusion* technique, which leverages the intermediate features of the generative models to jointly predict the noise for perturbed latent points and associated part segmentation labels during the denoising process, and subsequently decodes the latent points to point clouds conditioned on part segmentation labels. To effectively evaluate the quality of generated point clouds, we introduce a novel point cloud pairwise distance calculation method named *part-aware Chamfer distance* (p-CD). This method enables existing metrics, such as 1-NNA, to measure both the local structural quality and inter-part coherence of generated point clouds. Experiments on the large-scale synthetic dataset ShapeNet and real-world medical dataset IntrA, demonstrate that SeaLion achieves remarkable performance in generation quality and diversity, outperforming the existing state-of-the-art model, DiffFacto, by **13.33%** and **6.52%** on 1-NNA (p-CD) across the two datasets. Experimental analysis shows that SeaLion can be trained semi-supervised, thereby reducing the demand for labeling efforts. Lastly, we validate the applicability of SeaLion in generative data augmentation for training segmentation models and the capability of SeaLion to serve as a tool for part-aware 3D shape editing.

## 1 Introduction

In the past few years, 3D point cloud generation based on deep neural networks has attracted significant interest and achieved remarkable success in downstream tasks, such as 2D image to point cloud generation (Fan et al., 2017; Jiang et al., 2018) and point cloud completion (Yu et al., 2021; Huang et al., 2020). However, little effort has been devoted to the generative models capable of generating 3D point clouds with semantic segmentation labels. Exiting works (Gal et al., 2021; Li et al., 2022; Shu et al., 2019; Zhang et al., 2024) can generate point clouds composed of detachable sub-parts. Nevertheless, these sub-parts lack clear semantic meaning, hindering the application of generated point clouds in domains such as generative data augmentation for training segmentation models and semantic part-aware 3D shape editing.

Attributed to the effective approximation to the real data distribution, denoising diffusion probabilistic models (DDPMs) (Ho et al., 2020) outperform many other generative models such as variational autoencoders (VAEs) (Kingma & Welling, 2013) and generative adversarial networks (GANs) (Creswell et al., 2018) in generation quality and diversity. Current state-of-the-art diffusion-based point cloud generative models (Zeng et al., 2022; Luo & Hu, 2021; Zhou et al., 2021) have achieved impressive performance. However, they still lack the ability to generate semantic labels. To the best of our knowledge, DiffFacto (Nakayama et al., 2023) is the only recent work capable of generating point clouds with segmentation labels by utilizing multiple DDPMs to generate each part individually and predicting the pose of each part to assemble the entire point clouds. However, due to the part-wise generation factorization, DiffFacto exhibits limited part-to-part coherence within the generated shape.

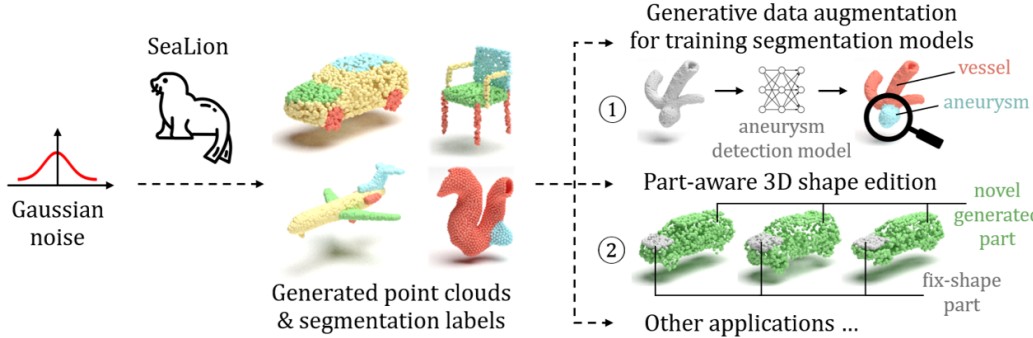

Figure 1: Leveraging the proposed semantic part-aware latent point diffusion technique, **SeaLion** generates high-quality point clouds with high inter-part coherence and accurate point-wise segmentation labels. The generated data has significant application potential, including enlarging the training sets for data-driven 3D segmentation models, particularly in medical examination domains where labeled data is scarce (①). Moreover, SeaLion can serve as a tool for part-aware 3D shape editing. ② shows examples of generated cars with varying shapes (green) and a fixed-shape hood (gray).

Inspired by (Baranchuk et al., 2021), which demonstrates that the intermediate hidden features learned by DDPMs can serve as representations capturing high-level semantic information for downstream vision tasks, we propose a novel approach that extends the generative model to not only predict noise for perturbed data but also point-wise part segmentation labels during the generation process. Our diffusion model learns the data distribution of regularized latent feature spaces, rather than directly approximating the distribution of point clouds in Euclidean space, since this latent diffusion strategy is proved to be more effective for complex point cloud generation (Zeng et al., 2022). Thereby, we train a VAE, conditioned by the segmentation labels, to map point clouds to latent points with point-wise segmentation awareness. During inference, the diffusion model simultaneously generates both latent points and their associated segmentation labels. The latter serves as conditional information for the VAE decoder, leading to generated point clouds with consistent segmentation labels. We refer this approach as semantic part-aware latent point diffusion technique. Based on it, we propose a generative model named SeaLion. Notably, the point-wise diffusion module in SeaLion utilizes a down-sampling data path to extract the common representations for both noise prediction and segmentation tasks, alongside two parallel up-sampling data paths to respectively extract task-specific features. SeaLion can generate high-quality point clouds with accurate segmentation labels. Besides, SeaLion diffuses on the latent points of all parts simultaneously, ensuring higher part-to-part coherence within a shape.

Currently, widely-used metrics for evaluating the quality of generated point clouds, such as 1-nearest neighbor accuracy (1-NNA) (Yang et al., 2019) and coverage (COV) (Achlioptas et al., 2018), fail to reflect the quality of segmentation-labeled point clouds. These metrics utilize Chamfer distance (CD) or earth mover's distance (EMD) (Rubner et al., 2000) to compute the pairwise point cloud distance, but neither of which considers the segmentation of point clouds. DiffFacto (Nakayama et al., 2023) uses the aforementioned metrics, but it assesses each part individually and then averages the results across all parts. However, this method still fails to measure the part-to-part coherence within a shape. We propose a novel evaluation metric named part-aware Chamfer distance (p-CD) to address these limitations and to quantify the pairwise distance between two segmentation-labeled point clouds. Using p-CD, evaluation metrics such as 1-NNA can effectively measure shape plausibility and part-to-part coherence of the generated point clouds.

We conduct extensive experiments on a large-scale synthetic dataset, ShapeNet (Yi et al., 2016), and a real-world 3D intracranial aneurysm dataset, IntrA (Yang et al., 2020). The results show that SeaLion achieves state-of-the-art performance in generating segmentation-labeled point clouds. Considering that labeling 3D point clouds is tedious, we evaluate SeaLion in a semi-supervised training setting, where only a small portion of the training data is labeled. Experimental results on ShapeNet validate that SeaLion can leverage additional unlabeled data, highlighting its potential to reduce labeling efforts. Ablation studies validate the feasibility of generative data augmentation using point clouds generated by SeaLion and the capability of SeaLion as a tool for part-aware 3D shape editing.

In summary, the contributions of this work are the following:

- We propose a novel generative model named SeaLion, capable of generating high-quality and diverse point clouds with accurate semantic segmentation labels.
- We propose a novel distance calculation method named part-aware Chamfer distance (p-CD), enabling widely-used metrics such as 1-NNA, COV, MMD to effectively evaluate the quality and diversity of segmentation-labeled point clouds.
- We demonstrate that SeaLion achieves state-of-the-art performance on a large synthesis dataset, ShapeNet, and a real-world medical dataset, IntrA. Furthermore, we show that SeaLion can be trained in a semi-supervised manner, reducing the need for labeling efforts.
- We confirm the feasibility of generative data augmentation using the point clouds generated by SeaLion and showcase SeaLion's function as a tool for part-aware 3D shape editing.

## 2 RELATED WORKS

**Detachable point cloud generation.** TreeGAN (Shu et al., 2019) conceptualizes point cloud generation as a tree growth process, where the final generated point cloud integrates various parts at the leaf nodes. SP-GAN (Li et al., 2021) maps a sphere in 3D space into a point cloud like FoldingNet (Yang et al., 2018), where different sphere regions correspond to other parts of the generated point cloud. MRGAN (Gal et al., 2021) achieves explicit part disentanglement by employing multiple branches of tree-structured graph convolution layers. EditVAE (Li et al., 2022) learns a disentangled latent representation for each part from point clouds in an unsupervised manner. Yet, the parts in these methods mentioned above do not necessarily possess clear semantic meaning. These methods are designed to facilitate the replacement of specific sub-parts for subsequent point cloud editing.

**Diffusion-based point cloud generation.** Point-Voxel Diffusion (Zhou et al., 2021) and DPM (Luo & Hu, 2021) train a diffusion model to generate point clouds directly. Instead, Lion (Zeng et al., 2022) utilizes a hierarchical VAE to map the point clouds to the global and point-level latent features and then trains latent diffusion models on them. Experimental results show that the latent diffusion model with a hierarchical encoding method can achieve better generation quality.

**Representations from generative models for discriminative tasks.** Some recent works explore using generative models as representation learners for discriminative tasks. (Donahue et al., 2016; Donahue & Simonyan, 2019; Chen et al., 2020) use the representations learned by GAN encoders and masked pixel predictors for 2D image classification. Without any additional training, (Li et al., 2023) chooses the category conditioning that best predicts the noise added to the input image as the classification prediction. (Zhang et al., 2021; Tritrong et al., 2021; Xu & Zheng, 2021; Xu et al., 2021; Baranchuk et al., 2021) investigate the usage of generative models on the segmentation tasks. (Baranchuk et al., 2021) shows that intermediate activations capture the semantic information from the input images and appear to be useful representations for the segmentation problem.

## 3 METHODOLOGY

In this section, we first give preliminaries on the diffusion models (Ho et al., 2020) and propose the *semantic part-aware latent points* technique. Next, we introduce the architecture of a novel generative model, *SeaLion*, and illustrate its usage as a part-aware 3D edition tool. Finally, we discuss the limitation of current metrics for evaluating generated labeled point clouds and propose novel metrics based on *part-aware Chamfer distance* (p-CD) that effectively measure shape plausibility and part-to-part coherence of the generated point clouds.

### 3.1 SEMANTIC PART-AWARE LATENT POINT DIFFUSION

The diffusion model (Ho et al., 2020) generates data by simulating a stochastic $T$-step process. During training, the diffusion model $\epsilon_\theta$ with parameters $\theta$ is trained to predict the noise $\epsilon$ to denoise the perturbed sample $x_t$ at step $t$. The training loss function is:

$$\mathcal{L}(\epsilon_\theta) = \mathbb{E}_{t,x_0,\epsilon}[||\epsilon_\theta(x_t, t, a) - \epsilon||_2^2], \qquad (1)$$

where $t \sim \text{Uniform}\{1, 2, ..., T\}$ is the diffusion time step, $\epsilon \sim \mathcal{N}(0, I)$ is the noise for diffusing $x_0$ to $x_t$, and $a$ is the conditional information, such as category encoding. During inference, the diffusion model starts from a random sample $x_T \sim \mathcal{N}(0, I)$ and denoises it iteratively until $t = 0$.

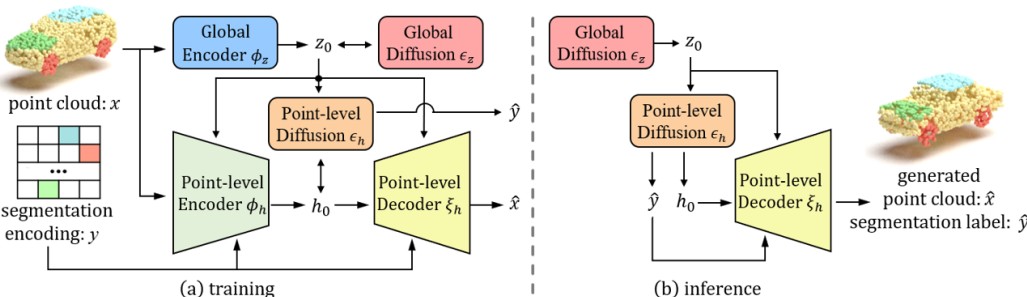

Figure 2: (a) **Training:** The point cloud $x$ is mapped to a global latent $z_0$ and semantic part-aware latent points $h_0$, with the segmentation encoding $y$ serving as the conditional information for $\phi_h$ and $\xi_h$. The diffusion $\epsilon_h$ is trained to denoise the perturbed latent points and predict the corresponding segmentation labels $\hat{y}$. (b) **Inference:** Starting from Gaussian noise, the diffusion modules generate $z_0$, $h_0$, and $\hat{y}$. Conditioning on $z_0$ and $\hat{y}$, the decoder $\xi_h$ produces a novel point cloud $\hat{x}$.

As a state-of-the-art model, Lion (Zeng et al., 2022) shows that mapping point clouds into regularized latent spaces and training DDPMs to learn the smoothed distributions is more effective than training DDPMs directly on complex point clouds. Given a point cloud $x \in \mathbb{R}^{n \times 3}$ consisting of $n$ points, Lion maps it to a global latent $z_0 \in \mathbb{R}^{d_z}$ and latent points $h_0 \in \mathbb{R}^{n \times d_h}$, and diffuses on these two latent features respectively. The $d_z$-dimensional vector $z_0$ encodes the global shape of the point cloud and serves as conditional information for point-level modules, while latent points $h_0$ encode the point-wise features and preserve the point cloud structure. However, the lack of semantic awareness of latent points hinders the generation of segmentation-labeled point clouds. Although the pseudo labels can be obtained by running a pre-trained segmentation model on the generated point clouds, this step-by-step method is vulnerable since it relies on an accurate segmentation model.

In 2D image domain, (Baranchuk et al., 2021) shows that intermediate latent features of DDPMs are informative for various computer vision tasks, thereby DDPMs can serve as powerful representation learners for tasks like image segmentation. Inspired by this insight, we propose *semantic part-aware latent point diffusion* technique for generating labeled point cloud. This technique builds on the hierarchical latent diffusion paradigm used in Lion but incorporates point-wise segmentation encodings $y$ as conditional information for the point-level encoder $\phi_h : \mathbb{R}^{n \times 3} \times \mathbb{R}^{n \times c} \times \mathbb{R}^{d_z} \rightarrow \mathbb{R}^{n \times d_h}$ and decoder $\xi_h : \mathbb{R}^{n \times d_h} \times \mathbb{R}^{n \times c} \times \mathbb{R}^{d_z} \rightarrow \mathbb{R}^{n \times 3}$, to obtain latent points $h_0$ with semantic part-awareness, as illustrated in Figure 2 (a). Given a point cloud $x$ and its associated segmentation encoding $y \in \mathbb{R}^{n \times c}$, the encoding and decoding process in the conditional VAE are as follows:

$$h_0 \leftarrow \phi_h(x, y, z_0), \quad \hat{x} \leftarrow \xi_h(h_0, y, z_0), \tag{2}$$

where $c$ is the number of segmentation parts, and $\hat{x}$ denotes the reconstructed point cloud that aligns with segmentation encoding $y$. This technique leverage the intermediate features of point-level diffusion model $\epsilon_h : \mathbb{R}^{n \times d_h} \times \mathbb{R} \times \mathbb{R}^{d_z} \rightarrow \mathbb{R}^{n \times d_h} \times \mathbb{R}^{n \times c}$ to predict the noise $\hat{\epsilon}_t$ for perturbed latent points $h_t$ and segmentation labels $\hat{y}_t$ at diffusion step $t$:

$$\hat{\epsilon}_t, \hat{y}_t \leftarrow \epsilon_h(h_t, t, z_0). \tag{3}$$

Over the denosing process, $\hat{y}_t$ is progressively smoothed to $\hat{y}$, which serves as conditional information for generating novel point clouds during inference, as illustrated in Figure 2 (b). To capture the features at different scales, we utilize a U-Net architecture in $\epsilon_h$. Notably, we use a down-sampling data path to extract common representations for both prediction tasks, alongside with two parallel up-sampling data paths for extracting task-specific features, as illustrated in Figure 3. Let $r_c$, $r_\epsilon$, and $r_y$ represent the intermediate features of representation learning, noise prediction, and segmentation prediction, respectively. Given the input $h_t$, the data flow in the down-sampling path is as follows:

$$r_c^i = \begin{cases} h_t, & i = 0, \\ f_c^i(r_c^{i-1}), & i \in \{1, ..., U\}, \end{cases} \tag{4}$$

where $f_c^i$ denotes the learnable encoding function at the $i$-th layer of down-sampling path, and $U$ represents the number of layers. For the noise prediction branch,

$$r_\epsilon^i = \begin{cases} f_\epsilon^i(r_c^i), & i = U, \\ f_\epsilon^i(r_\epsilon^{i+1} \oplus r_c^i), & i \in \{U-1, ..., 0\}, \end{cases} \tag{5}$$

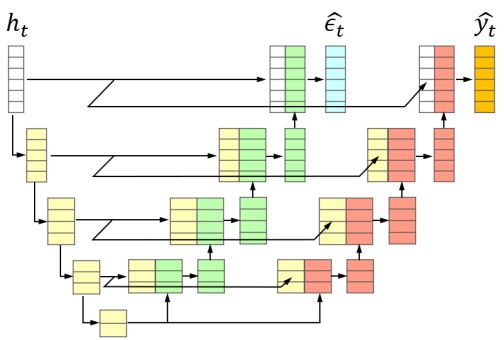

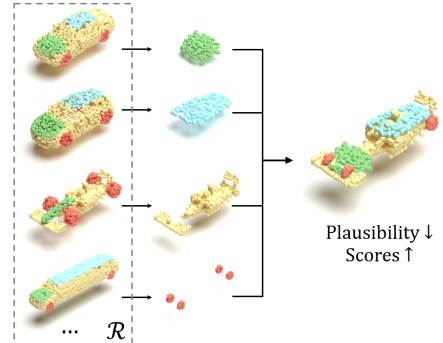

Figure 3: Data flow in the point-level diffusion module $\epsilon_h$. The input, perturbed latent points $h_t$ at step $t$, is down-sampled and transformed to representations $r_c$ (yellow). Two parallel up-sampling paths concatenate $r_c$ with task-specific features, $r_\epsilon$ (green) and $r_y$ (red), to separately predict the noise $\hat{\epsilon}_t$ and the segmentation encoding $\hat{y}_t$.

Figure 4: Limitations of the intra-part and inter-part scores. By combining parts from the real dataset $\mathcal{R}$ and maintaining the connection tightness, we can generate a set of implausible samples that still achieves high scores on both metrics.

where $f_\epsilon^i$ denotes the learnable encoding function at the $i$-th layer of noise prediction branch, and $\oplus$ is the concatenation operation. The same paradigm applies to the segmentation prediction branch. The final output of $\epsilon_h$ are the predicted noise and segmentation labels, i.e. $\hat{\epsilon}_t \leftarrow r_\epsilon^0$ and $\hat{y}_t \leftarrow r_y^0$. Besides, this technique involves a global encoder $\phi_z : \mathbb{R}^{n \times 3} \rightarrow \mathbb{R}^{d_z}$ and a diffusion module $\epsilon_z : \mathbb{R}^{d_z} \times \mathbb{R} \rightarrow \mathbb{R}^{d_z}$ for the global latent $z_0$, which encodes the overall information of the point cloud.

**Training.** Using this technique, the training consists of two stages. In the first stage, we train the components of hierarchical VAE, including $\phi_z$, $\phi_h$, and $\xi_h$, to maximize a variational lower bound on the data log-likelihood (ELBO):

$$\begin{aligned}
\mathcal{L}(\phi_z, \phi_h, \xi_h) = \mathbb{E}_{p(x), q_{\phi_z}(z_0|x), q_{\phi_h}(h_0|x,y,z_0)} \{\log p_{\xi_h}(x|h_0, y, z_0) \\
- \lambda_z D_{KL}[q_{\phi_z}(z_0|x)|\mathcal{N}(0,I)] - \lambda_h D_{KL}[q_{\phi_h}(h_0|x,y,z_0)|\mathcal{N}(0,I)]\},
\end{aligned} \tag{6}$$

where $q_{\phi_z}$ and $q_{\phi_h}$ are the posterior distribution for sampling $z_0$ and $h_0$, $p_{\xi_h}$ is the prior for reconstruction prediction, and $\lambda_z$ and $\lambda_h$ are the hyperparameters for balancing reconstruction accuracy and Kullback-Leibler regularization. In the second stage, we train two diffusion modules $\epsilon_z$ and $\epsilon_h$. The objectives for $\epsilon_z$ and $\epsilon_h$ are as follows:

$$\mathcal{L}(\epsilon_z) = \mathbb{E}_{t,z_0,\epsilon}[||\epsilon_z(z_t,t) - \epsilon||_2^2], \tag{7}$$

$$\mathcal{L}(\epsilon_h) = \mathbb{E}_{t,h_0,\epsilon}[||\hat{\epsilon}_t - \epsilon||_2^2 + \lambda_{seg} H(y, \hat{y}_t)], \tag{8}$$

where $\epsilon \sim \mathcal{N}(0,I)$ denotes the added noise, $H(\cdot)$ is cross entropy, and $\lambda_{seg}$ is the hyperparameter for balancing two prediction tasks.

**Inference.** As illustrated in Figure 2 (b), the inference process consists of three steps. The global diffusion $\epsilon_z$ firstly generates a global latent $z_0$. Conditioning on $z_0$, the point-level diffusion $\epsilon_h$ then generates the latent points $h_0$ and the associated segmentation prediction $\hat{y}$. Since SeaLion predicts segmentation $\hat{y}_t$ at each denoising step, we apply an exponential moving average (EMA) to smooth the prediction results. This process is repeated for each backward diffusion step $t$, from $T$ to 0,

$$\overline{y}_t = \begin{cases} \hat{y}_t, & t = T, \\ \alpha \, \hat{y}_t + (1 - \alpha) \, \overline{y}_{t+1}, & t < T, \end{cases} \tag{9}$$

where $\overline{y}_t$ is the smoothed segmentation prediction result at step $t$, and $\alpha$ is the smoothing factor of EMA. We take $\overline{y}_0$ as the final prediction result $\hat{y}$. Lastly, conditioning on $\hat{y}$ and $z_0$, the point-level decoder $\xi_h$ transforms $h_0$ to the generated point cloud $\hat{x}$.

### 3.2 MODEL ARCHITECTURE OF SEALION

Based on the semantic part-aware latent point diffusion technique, we introduce a novel point cloud generative model named SeaLion. The architecture of SeaLion is illustrated as follows:

**Point-level encoder $\phi_h$ and decoder $\xi_h$.** In SeaLion, $\phi_h$ and $\xi_h$ adopt a similar 4-layer Point-Voxel CNN (PVCNN) (Liu et al., 2019) as their backbones. PVCNN, a U-Net style architecture for point cloud data, uses set abstraction layer (Qi et al., 2017) and feature propagation layer (Qi et al., 2017) for down-sampling and up-sampling the points. Point-voxel convolutions (PVConv) blocks (Liu et al., 2019), which merge the advantages of point-based and voxel-based methods, are utilized to extract neighboring features at each layer. To incorporate the conditional information, the global latent $z_0$ is integrated through the adaptive Group Normalization (Zeng et al., 2022) in PVConv, while the segmentation encoding $y$ is concatenated with the intermediate features at each layer.

**Point-level diffusion $\epsilon_h$.** As discussed in 3.1, point-level diffusion $\epsilon_h$ contains a down-sampling path to learn the shared representations for both prediction task and two parallel up-sampling paths to extract the task-specific features. Accordingly, we adopt a modified PVCNN architecture with one down-sampling path and two up-sampling branches.

**Global encoder $\phi_z$ and diffusion $\epsilon_z$.** We adopt the same architectures as Lion for the two global-level modules. The global encoder $\phi_z$ consists of PVConv blocks, set abstraction layers, a max pooling layer, and a multi-layer perceptron. The global diffusion $\epsilon_z$ comprises stacked ResNet (He et al., 2016). More details regarding the model architecture can be found in the supplementary materials.

### 3.3 PART-AWARE 3D SHAPE EDITION TOOL

Since the latent points are semantic part-aware, SeaLion can be used as a part-aware 3D shape edition tool. Given a point cloud $x$ consisting of $|P|$ parts, where we aim to preserve part $p \in P$ while introducing variations to the remaining parts. After transforming the point cloud to the latent points $h$, we can freeze the latent points belonging to part $p$ and apply the diffusion-denoise process (Zeng et al., 2022; Meng et al., 2022) on the unfrozen latent points. During this process, the unfrozen latent points are perturbed for $\tau$ steps ($\tau < T$) and then denoised recursively for the same number of steps. Due to the stochasticity of the denoising process, the unfrozen latent points will differ after denosing, leading to deformations in the corresponding parts when decoded by $\xi_h$. The pseudo code of using SeaLion as an editing tool is provided in the supplementary materials.

### 3.4 EVALUATION METRICS

**Notions.** Given a generated dataset $\mathcal{G} = \{x^g | x^g \in \mathbb{R}^{n \times 3}\}$ and a real dataset $\mathcal{R} = \{x^r | x^r \in \mathbb{R}^{n \times 3}\}$, both consist of point clouds with $n$ points. Suppose each point cloud $x \in \mathbb{R}^{n \times 3}$ consists of $|P|$ parts, i.e. $x = \{x_p | p \in P, x_p \in \mathbb{R}^{n_p \times 3}\}$, where $n_p$ is the number of points in part $p$. For example, if $x$ represents a car from ShapeNet (Yi et al., 2016), $P = \{\text{roof, hood, wheels, body}\}$.

**Existing metrics.** The essential of evaluating point cloud generation is to assess both the quality and diversity of the generated data. Most existing works (Zeng et al., 2022; Zhou et al., 2021; Yang et al., 2019) use metrics such as 1-nearest neighbor accuracy (1-NNA) (Yang et al., 2019), coverage (COV), and minimum matching distance (MMD) (Achlioptas et al., 2018) for evaluation. The formulas for these metrics are provided in the supplementary materials. As discussed in (Yang et al., 2019; Zeng et al., 2022), COV quantifies generation diversity and is sensitive to mode collapse, but it fails to evaluate the quality of $\mathcal{G}$. MMD, on the other hand, only assess the best quality point clouds in $\mathcal{G}$ and is not a reliable metric to measure overall generation quality and diversity. 1-NNA (Yang et al., 2019) measures both generation quality and diversity by quantifying the distribution similarity between $\mathcal{R}$ and $\mathcal{G}$. The aforementioned metrics rely on Chamfer distance (CD) or earth mover's distance (EMD) (Rubner et al., 2000) to measure the distance between two point clouds. However, neither CD nor EMD considers the semantic segmentation of the points, making these metrics ineffective to evaluate the generated point clouds with point-wise segmentation labels.

DiffFacto (Nakayama et al., 2023) introduces *intra-part* and *inter-part scores* to evaluate the quality of segmentation-labeled point clouds. The intra-part score measures the quality of the independently generated parts and the overall point cloud by averaging the results across all parts. For example, 1-NNA-P (Nakayama et al., 2023) is the average of 1-NNA score for all parts, computed as follows:

$$\text{1-NNA-P}(\mathcal{R}, \mathcal{G}) = \frac{1}{|P|} \sum_{p \in P} \frac{\sum_{x_p^r \in \mathcal{R}_p} \mathbb{1}[N_{x_p^r} \in \mathcal{R}_p] + \sum_{x_p^g \in \mathcal{G}_p} \mathbb{1}[N_{x_p^g} \in \mathcal{G}_p]}{|\mathcal{R}_p| + |\mathcal{G}_p|}, \tag{10}$$

where $\mathcal{G}_p := \{x_p^g\}$ and $\mathcal{R}_p := \{x_p^r\}$ represent the generated and real sets of part $p$, respectively, and $\mathbb{1}[\cdot]$ is the indicator function. $N_{x_p^r}$ is the nearest neighbor of $x_p^r$ in the set $\mathcal{R}_p \cup \mathcal{G}_p \setminus \{x_p^r\}$, with the same applying to $N_{x_p^g}$. The nearest neighbor is determined according to the Chamfer distance. Given two parts $x_p^1$ and $x_p^2$, the Chamfer distance between them is computed by:

$$\text{Chamfer}(x_p^1, x_p^2) = \frac{1}{|x_p^1|} \sum_{q_1 \in x_p^1} \min_{q_2 \in x_p^2} ||q_1 - q_2||_2^2 + \frac{1}{|x_p^2|} \sum_{q_2 \in x_p^2} \min_{q_1 \in x_p^1} ||q_1 - q_2||_2^2, \quad (11)$$

where $q_1, q_2 \in \mathbb{R}^3$ represent points belonging to parts $x_p^1, x_p^2$. The inter-part score, the snapping metric (SNAP) (Nakayama et al., 2023), measures the connection tightness between two contacting parts in a object. The formula for SNAP is provided in the supplementary materials. However, both intra-part and inter-part scores have limitations in evaluating the generation of segmentation-labeled point clouds. Specially, averaging the score among all parts or measuring the connection tightness does not effectively measures the coherence among the parts within an object. An extreme case is illustrated in Figure 4. By recombining parts from different shapes in the real dataset and maintaining connection tightness, we can create a generated set of implausible samples that still archives high score on the aforementioned metrics.

**Part-aware Chamfer distance.** To address this issue, we propose part-aware Chamfer Distance (p-CD). Given point clouds $x^1$ and $x^2$ consisting of $P$ parts, the pairwise distance is calculated by:

$$\text{p-CD}(x^1, x^2) = \sum_{p \in P} \left\{ \frac{1}{|x_p^1|} \sum_{q_1 \in x_p^1} \min_{q_2 \in x_p^2} ||q_1 - q_2||_2^2 + \frac{1}{|x_p^2|} \sum_{q_2 \in x_p^2} \min_{q_1 \in x_p^1} ||q_1 - q_2||_2^2 \right\}. \quad (12)$$

In p-CD, all parts of the point clouds are taken into account. Therefore, if a generated point cloud has a small p-CD to a real point cloud, it indicates that not only are all parts of the generated point cloud of high quality, but they also form a coherent and reasonable assembly as a whole. Consequently, the randomly assembled sample in Figure 4 will have a large p-CD to the real samples, indicating the anomaly of the generated sample. Based on p-CD, we can compute the 1-NNA (p-CD), COV (p-CD), and MMD (p-CD) to measure the part-aware proximity of a generated set to a real set.

## 4 EXPERIMENTS

In this section, we first describe the experimental setup, including the datasets, training details and evaluation metrics. Next, we present the evaluation results and the generated point clouds of SeaLion on ShapeNet (Yi et al., 2016) and IntrA (Yang et al., 2020). In the experimental analysis, we demonstrate that SeaLion can be trained in a semi-supervised manner, reducing the reliance on labeled data. Furthermore, we showcase the applicability of SeaLion for generative data augmentation in the point cloud segmentation task and SeaLion's function as a tool for part-aware shape editing.

### 4.1 EXPERIMENTAL SETUP

**Datasets.** We conduct experiments on two public datasets, ShapeNet (Yi et al., 2016) and IntrA (Yang et al., 2020). ShapeNet (Yi et al., 2016) is a large-scale synthetic dataset of 3D shapes with semantic segmentation labels. We use six categories from this dataset: airplane, car, chair, guitar, lamp, and table. SeaLion is trained and tested for each category using the official split. IntrA (Yang et al., 2020) is a real-world dataset containing 3D intracranial aneurysm point clouds reconstructed from MRI. The dataset contains 116 aneurysm segments manually annotated by medical experts. We randomly select 93 segments for training and use the remaining 23 segments for testing. Each aneurysm segment includes the healthy vessel part and the aneurysm part.

**Training Details.** As discussed in 3.1, the training of SeaLion includes two stages. In this work, we train the VAE model for 8k epochs in the first stage and the latent diffusion model for 24k epochs in the second stage. For these two stages, we use an Adam optimizer with a learning rate of 1e-3.

**Metrics.** We use the part-aware Chamfer distance (p-CD) proposed in 3.4 to quantify the pairwise point cloud distance. As discussed in (Yang et al., 2019), 1-NNA can measure both generation quality and diversity by computing the distribution similarity between $\mathcal{R}$ and $\mathcal{G}$, while COV and MMD have limitations in measuring the overall generation quality. Therefore we compute 1-NNA (p-CD)

| Metric | Model | Airplane | Car | Chair | Guitar | Lamp | Table |
|---|---|---|---|---|---|---|---|
| | Lion & PointNet++ | 68.48 | 79.11 | 65.42 | - | - | - |
| 1-NNA (p-CD) ↓ (%) | DiffFacto | 81.67 | 90.51 | 77.34 | - | 67.13 | - |
| | **SeaLion** | **65.40** | **73.10** | **63.14** | **62.59** | **61.71** | **63.56** |
| | Lion & PointNet++ | 39.00 | 33.54 | 43.75 | - | - | - |
| COV (p-CD) ↑ (%) | DiffFacto | 32.26 | 26.58 | 35.37 | - | 46.95 | - |
| | **SeaLion** | **47.51** | **44.94** | **46.88** | **46.85** | **48.25** | **41.04** |
| | Lion & PointNet++ | **5.91** | 8.18 | 17.13 | - | - | - |
| MMD (p-CD) ↓ ($\times 10^{-3}$) | DiffFacto | 7.15 | 9.03 | 20.30 | - | 29.47 | - |
| | **SeaLion** | 6.38 | **7.95** | **16.25** | **2.11** | **28.38** | **14.56** |

Table 1: Evaluation on ShapeNet (Yi et al., 2016). Note that certain data is missing because DiffFacto (Nakayama et al., 2023) only provides pretrained models for airplane, car, chair, and lamp categories, while Lion (Zeng et al., 2022) only releases generated point clouds for airplane, car, and chair categories.

as the primary evaluation metric in this work, but we still report COV (p-CD), and MMD (p-CD) for convenience of other researchers. Additionally, we report the results of 1-NNA-P, COV-P, and MMD-P (Nakayama et al., 2023) for the airplane and chair categories in ShapeNet for a comparison to DiffFacto, despite the limitation of these metrics has been discussed in 3.4.

## 4.2 EXPERIMENTAL RESULTS

**Evaluation on ShapeNet.** The experiment results of SeaLion on the six classes in ShapeNet are presented in Table 1. DiffFacto (Nakayama et al., 2023) provides pretrained weights for four categories in ShapeNet: airplane, car, chair, and lamp. We use these released weights to generate point clouds and evaluate them using our proposed metrics. Additionally, we use a pretrained PointNet++ (Qi et al., 2017) to generate segmentation labels for the officially released point clouds generated from Lion (Zeng et al., 2022). The results show that SeaLion outperforms DiffFacto and the hybrid generation method combining Lion and PointNet++. For the airplane, car, chair, and lamp categories, SeaLion outperforms DiffFacto by an average of **13.33%** on 1-NNA (p-CD), **11.61%** on COV (p-CD), and **10.60%** on MMD (p-CD), indicating that SeaLion generates higher-quality and more diverse data. Some of the generated point clouds are demonstrated in Figure 5, showing not only plausible shape and part-to-part coherence but also high variety among the shapes. More generated point clouds are provided in the supplementary materials. Besides, we report the evaluation of SeaLion according to 1-NNA-P, COV-P, and MMD-P (Nakayama et al., 2023) in Table 2. The results show that SeaLion outperforms DiffFacto on the primary metric 1-NNA-P and achieves competitive performance on the other metrics. By comparing the results of DiffFacto (Nakayama et al., 2023) in Table 1 and Table 2, we can observe a notable drop from 1-NNA-P to 1-NNA (p-CD), indicating that 1-NNA-P does not effectively capture the part-to-part coherence in the generated shapes, whereas 1-NNA (p-CD) provides a more reliable evaluation of shape consistency.

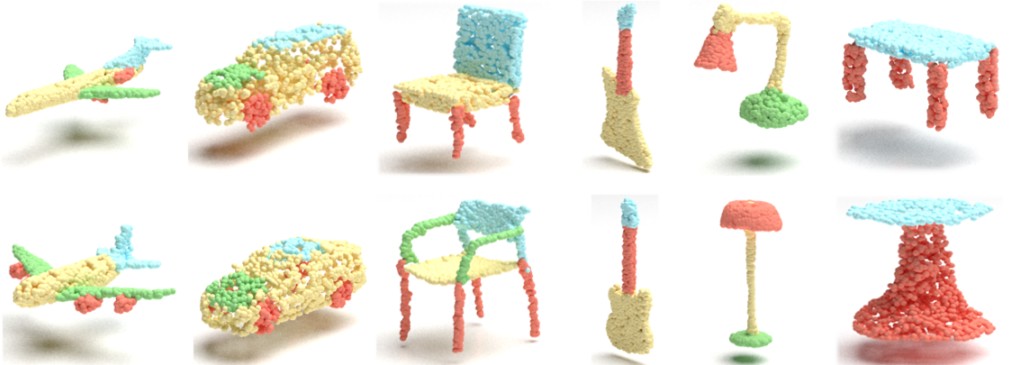

Figure 5: Generated point clouds of airplanes, cars, chairs, guitars, lamps and tables from SeaLion.

| Metric | Model | Airplane | Chair |
|---|---|---|---|
| 1-NNA-P ↓ (%) | Lion & PointNet++ | 68.73 | 69.25 |
| | DiffFacto | 68.72 | 65.23 |
| | **SeaLion** | **68.39** | **63.24** |
| COV-P ↑ (%) | Lion & PointNet++ | 38.8 | 35.1 |
| | DiffFacto | **46.2** | 42.5 |
| | **SeaLion** | 44.9 | **46.5** |
| MMD-P ↓ $(\times 10^{-2})$ | Lion & PointNet++ | 3.68 | 3.99 |
| | DiffFacto | **3.20** | 3.27 |
| | **SeaLion** | 3.45 | **2.73** |

Table 2: Evaluation on airplane and chair classes in ShapeNet (Yi et al., 2016) according to the metrics proposed in DiffFacto (Nakayama et al., 2023).

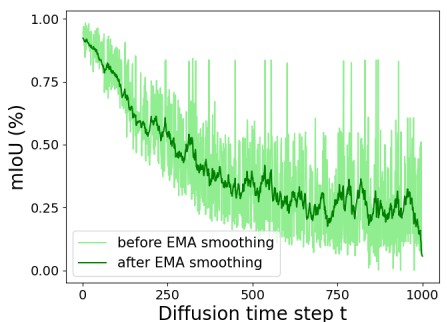

Figure 6: Evolution of predictive performance measured by mIOU for different diffusion steps on airplane class.

In SeaLion, the diffusion $\epsilon_h$ predicts both noise and segmentation during the generation process. We demonstrate the evolution of predictive performance, measured by mIoU, across different diffusion steps $t$ for the airplane category in Figure 6. As $t$ decreases from $T$ to 0 during the denosing process, the perturbed latent points $h_t$ become increasingly informative for segmentation prediction. This trend aligns with the findings in (Baranchuk et al., 2021).

**Evaluation on IntrA.** In this experiment, we train both SeaLion and DiffFacto (Nakayama et al., 2023) on on the IntrA dataset (Yang et al., 2020) for comparison. The experimental results presented in Table 3 demonstrate that SeaLion outperforms DiffFacto by **6.52%** on 1-NNA (p-CD), **21.74%** on COV (p-CD), and **8.45%** on MMD (p-CD). Some of the generated intracranial aneurysm point clouds from SeaLion are presented in Figure 7.

### 4.3 EXPERIMENTAL ANALYSIS

Compared with collecting 3D data, which can be automated using tools like web crawlers, manually labeling segmentation is tedious and time-consuming. Therefore, method for extracting the information from the unlabeled data attracts lots of attention in recent years. Typically, semi-supervised learning effectively reduces the need for extensive data labeling by training models with a combination of a small amount of labeled samples and a larger set of unlabeled samples. The training process of DiffFacto (Nakayama et al., 2023) involves separate training for each semantic part, which limits its ability to leverage the unsegmented data. In contrast, SeaLion generates the points for all parts jointly, making it adaptable to the semi-supervised training approach. Given an unlabeled sample, we can replace the segmentation encoding $y$ in equation 6 with zero padding of the same shape, thereby transforming the corresponding modules to be unconditioned by $y$. Additionally, we omit the second term $H(y, \hat{y}_t)$ in equation 8 to skip the training of segmentation prediction on unsegmented samples. Consequently, SeaLion can be trained on unlabeled samples using this approach, while labeled samples can still be processed using the objective functions in 3.1.

To validate the applicability of SeaLion trained using a semi-supervised approach, we conduct an experiment on the car class in ShapeNet (Yi et al., 2016). We randomly select 10% of the samples in

| Metric | Model | Aneurysm |
|---|---|---|
| 1-NNA (p-CD) ↓ (%) | DiffFacto | 71.74 |
| | **SeaLion** | **65.22** |
| COV (p-CD) ↑ (%) | DiffFacto | 39.13 |
| | **SeaLion** | **60.87** |
| MMD (p-CD) ↓ $(\times 10^{-2})$ | DiffFacto | 8.05 |
| | **SeaLion** | **7.37** |

Table 3: Evaluation on IntrA (Yang et al., 2020).

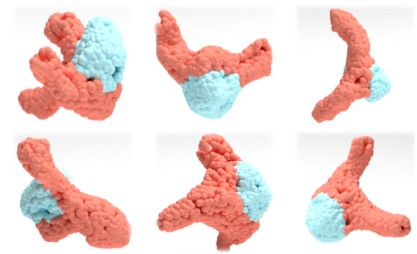

Figure 7: Generated aneurysm segments from SeaLion (red: vessels, blue: aneurysm).

| Metric | Model | Training Set | Car |
|---|---|---|---|
| 1-NNA (p-CD) ↓ (%) | DiffFacto | $\mathcal{L}$ | 90.82 |
| | SeaLion | $\mathcal{L}$ | 87.34 |
| | **SeaLion** | $\mathcal{L} + \mathcal{U}$ | **83.23** |
| COV (p-CD) ↑ (%) | DiffFacto | $\mathcal{L}$ | 23.42 |
| | SeaLion | $\mathcal{L}$ | 37.34 |
| | **SeaLion** | $\mathcal{L} + \mathcal{U}$ | **41.77** |
| MMD (p-CD) ↓ ($\times 10^{-3}$) | DiffFacto | $\mathcal{L}$ | 9.37 |
| | SeaLion | $\mathcal{L}$ | 8.76 |
| | **SeaLion** | $\mathcal{L} + \mathcal{U}$ | **8.33** |

Table 4: Evaluation of the semi-supervised training on SeaLion. In the training set column, $\mathcal{L}$ refers to the use of 10% data with segmentation labels, while $\mathcal{U}$ refers to the remaining data without segmentation labels.

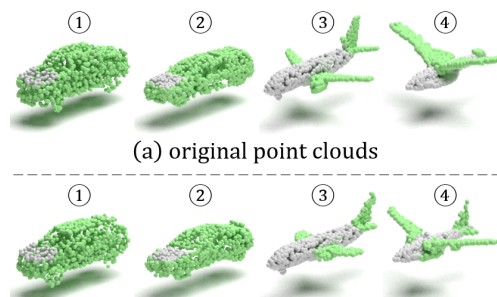

(a) original point clouds

(b) novel generated point clouds

Figure 8: (a) Original point clouds and (b) novel generated point clouds after part-aware editing (gray: fix-shape parts, green: novel generated parts with deformations).

the training set as labeled data, while the remaining 90% are treated as unsegmented. For comparison, we train three models as follows: (1) DiffFacto trained with 10% labeled data using the official default settings, (2) SeaLion trained with 10% labeled data, and (3) SeaLion trained in a semi-supervised approach with 10% labeled data and 90% unlabeled data. The experimental results presented in Table 4 demonstrate that SeaLion outperforms DiffFacto when trained with 10% labeled data, and its performance further improves after incorporating unlabeled data into the training set.

## 4.4 APPLICATIONS

**Generative data augmentation.** In this experiment, we use the point clouds generated by SeaLion to enlarge the dataset for training the data-driven segmentation model. We use SPoTr (Park et al., 2023), a state-of-the-art and open source model on ShapeNet segmentation benchmark, to predict the part segmentation across six categories in ShapeNet. We evaluate the performance of SPoTr using the mIoU metric. The results, presented in Table 5, demonstrate that the incorporation of generative data steadily enhances the performance of SPoTr across all categories.

**Part-aware 3D shape edition.** As discussed in 3.3, SeaLion can serve as a tool for part-aware point cloud edition tool by running the diffusion-denoise process on the latent points associated with the parts the user wishes to modify. We conduct experiments on car and airplane point clouds, where the hood of cars and the body of airplanes are set as the fixed-shape parts. The experimental results illustrated in Figure 8 shows that novel generated cars and airplanes keep the chosen parts (hoods and airplane cabins) unchanged while exhibiting diverse deformation in the remaining parts.

| Model | Training Set | Airplane | Car | Chair | Guitar | Lamp | Table |
|---|---|---|---|---|---|---|---|
| SPoTr | real labeled data | 82.28 | 76.98 | 90.31 | 90.97 | 82.50 | 82.77 |
| **SPoTr** | real & generated labeled data | **83.81** | **79.43** | **90.88** | **91.56** | **84.54** | **83.44** |

Table 5: Generative data augmentation for training SPoTr (Park et al., 2023).

## 5 CONCLUSION

In this paper, we introduce the semantic part-aware latent point diffusion technique for generating point clouds with segmentation labels. Using this technique, our novel generative model, SeaLion, achieves state-of-the-art performance on ShapeNet and IntrA datasets. Additionally, we discuss the limitations of the existing metrics for evaluating the generated labeled point clouds and propose better metrics based on a novel point cloud pairwise distance calculation method named part-aware Chamfer distance. Further experiments validate the feasibility of generative data augmentation using the point clouds generated by SeaLion and the utility of SeaLion as a tool for part-aware 3D shape editing, highlighting the broad applicability of SeaLion in various downstream tasks.

## REPRODUCIBILITY STATEMENT

The hyper-parameters for implementing SeaLion are provided in Tables 1-5 of the supplementary materials. The mathematical formulas for the metrics used in this paper are provided in Section 3.4 of the main paper and Section 2 of the supplementary materials. The pseudo code for using SeaLion as a part-aware 3D shape editing tool is provided in Section 1 of the supplementary materials.

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
