# OpenReview forum: "SeaLion: Semantic Part-Aware Latent Point Diffusion Models for 3D Generation"
_ICLR.cc/2025/Conference — ICLR 2025 Conference Withdrawn Submission_

### Official Review · Reviewer_j97B · 2024-10-22

**Soundness:** 2
**Presentation:** 2
**Contribution:** 2
**Rating:** 5
**Confidence:** 4

**Summary:**

This paper introduces a semantic part-aware latent point diffusion technique for generating point clouds with segmentation labels, which achieves state-of-the-art performance on ShapeNet and IntrA datasets. Additionally, this paper also discusses the limitations of the existing metrics for evaluating the generated labeled point clouds and propose better metrics based on a novel point cloud pairwise distance calculation method named part-aware Chamfer distance.

**Strengths:**

This paper proposes a relatively novel task, using diffusion models to simultaneously generate 3D shapes and segmentation labels. The proposed semantic-aware latent point diffusion technique is indeed an innovative idea, addressing the lack of labeled data and dependency on pre-trained segmentation models in a semi-supervised manner.

**Weaknesses:**

I have the following concerns:
1)	Is it necessary to have a single metric that evaluates both generation quality and segmentation quality? Why not assess generation quality and segmentation accuracy separately? This would seem more convincing for both tasks.
2)	The paper proposes a "semantic part-aware latent point diffusion" but fails to clarify how this is achieved, only vaguely mentioning that segmentation labels are incorporated into the encoder.
3)	In line 414 of the paper, it is claimed that comparing 1-NNA-P and 1-NNA(p-CD) leads to the conclusion that the latter provides better shape consistency. Did you use this evaluation metric as a training loss? Is there any qualitative comparison to support your conclusion?
4)	During 3D shape editing, it seems that only part of the points can be fixed while the other part is randomly generated. This kind of "editing" operation seems to have limited significance.

**Questions:**

Same to weaknesses

---

### Official Review · Reviewer_ESoL · 2024-11-01

**Soundness:** 3
**Presentation:** 3
**Contribution:** 3
**Rating:** 6
**Confidence:** 3

**Summary:**

1. **Overall**: SeaLion introduces a method for semantic-aware point cloud generation using diffusion models, marking an early exploration in this field alongside Diffacto.

2. **Method and Performance**: Building on the Lion model, SeaLion adds a new branch that leverages the latent features of the diffusion model for segmentation learning to achieve part-aware generative capability. Experiments show it achieves good quantitative performance compared to Diffacto and Lion+Segmentation.

3. **p-CD**: The paper also discusses evaluation metrics for this task, highlighting limitations in the metrics proposed by Diffacto. SeaLion introduces a new metric, `p-DC`, that accounts for the structural integration of parts rather than evaluating each part individually, + tidiness calculations like Diffacto.

4. **Application**: Finally, SeaLion is tested in a part-aware 3D editing application, enabling selective control over a 3D model by keeping specific parts fixed while allowing others to be generatively modified.

**Strengths:**

- The proposed task is important
- The paper is well-written.
- The proposed method performs well, with thorough evaluation conducted on two datasets.
- The discussion of evaluation metrics is logical, and the newly proposed metric is reasonable (although some questions remain).

**Weaknesses:**

1. The newly proposed metric seems to calculate only part alignment between objects, but it does not incorporate overall alignment. Perhaps using it as part of the distance calculation, weighted with the original distance function, would make more sense. Intuitively, it's likely that each part of two objects aligns well individually, but the overall structure of generative object are bad. I would like to hear more discussion on this point.

2. I understand that the focus of the task is on semantic-aware 3D generation, but it would be interesting to see whether adding the branch in the diffusion model improves or reduces the overall point cloud quality. In other words, evaluating performance based on the entire structure, rather than solely on individual parts, could reveal whether incorporating component information in the final generated point cloud subtly impacts overall coherence.

3. Additional visualizations demonstrating how the introduced metric reflect alignment would be helpful.

**Questions:**

1. In the case where generated point cloud lacks some semantic parts present in the real point cloud, How is pCD calculated? For example, if it generates a chair with only a backrest and seat, but no arms, how is the distance evaluated against a chair that has all three parts? Referring to Figures 2 and 3 in the supplementary material, green parts are not present in all generated data. (This issue does not occur with the original metric proposed by Diffacto.)

2. Regarding the training scheme, how is training performed? Is the model trained on multiple classes so that the generative model can generate multiple objects? If so, similar to question 1, how is p-CD calculated when comparing, say, a car and a chair?

3. It would be interesting to know if adding the new branch for segmentation improves results compared to LION when evaluated on overall performance rather than segmentation.

---

### Official Review · Reviewer_q99a · 2024-11-02

**Soundness:** 3
**Presentation:** 3
**Contribution:** 2
**Rating:** 6
**Confidence:** 4

**Summary:**

This paper studies generating point clouds with point-wise segmentation labels. To achieve it, this paper proposes the semantic part-aware latent point diffusion technique which can  jointly predict the noise for perturbed latent points and associated part segmentation labels during the denoising process.

In addition, to evaluate the quality of generated point clouds, this paper further proposes part-aware Chamfer distance that enables existing metrics to measure both the local structural quality and inter-part coherence of generated point clouds.

To validate the effectiveness, this paper conducts experiments on ShapeNet and IntrA. The proposed method achieves better scores than previous works.

**Strengths:**

- This paper is well-written and easy to follow. The motivation is also clear.

- The studied task that is under-explored is interesting.

- The performance is good.

**Weaknesses:**

- The proposed method seems to only generate the point cloud the class of which exists in the training dataset. So this method can be considered as a data augmentation technique and the authors show its value in Table 5. However, this experiment is not enough and the increase in performance is not significant. Do the authors compare it with other data augmentations?

- It seems that we cannot control the class of the generated point cloud, so how to guarantee the class balance of the generated point cloud data.

- In Table 1, the authors only use PointNet++ to generate the pseudo label of generated data using Lion. Why do not use latest or SOTA methods, like SPoTr.

- Regarding the evaluation for the accuracy of the semantic labels, do the authors have any idea?

**Questions:**

Please give explanations for the question in weaknesses

---

### Official Review · Reviewer_gV59 · 2024-11-03

**Soundness:** 2
**Presentation:** 3
**Contribution:** 1
**Rating:** 3
**Confidence:** 5

**Summary:**

The paper presents SeaLion, a semantic part-aware latent point diffusion model that aims to simultaneously generate 3D point clouds along with semantic segmentation labels. The authors introduce a part-aware latent point diffusion approach that leverages intermediate latent features to predict both the 3D structure and the segmentation labels during the generation process. To evaluate the generated point clouds, the paper proposes a part-aware Chamfer distance (p-CD), designed to assess both local structure and inter-part coherence. Experiments on ShapeNet and IntrA datasets claim that SeaLion surpasses existing models in both quality and consistency of generated segmentation-labeled point clouds.

**Strengths:**

The authors attempt to address an interesting challenge by introducing a joint 3D shape and segmentation generation approach, and they propose a new evaluation metric (p-CD) to quantify coherence across parts within generated point clouds. However, these strengths are limited by the lack of sufficient motivation and empirical analysis supporting the proposed methodology.

**Weaknesses:**

1.	Unclear Motivation for Joint Generation: The choice to couple the generation of 3D shapes with segmentation labels lacks clear motivation. The paper does not address why the segmentation cannot be applied as a post-processing step using a state-of-the-art segmentation model, which could yield similar or better results without added complexity.
	2.	Probabilistic Mismatch: Since 3D points and segmentation labels may reside in different probability distributions, it is questionable whether combining them within a single generative model is theoretically sound. This approach could complicate the training process without a demonstrable improvement in quality or coherence.
	3.	Alternative Approaches Ignored: The authors do not explore simpler alternatives, such as applying segmentation using an additional classification head after generating the 3D point clouds. This simpler approach could potentially achieve high-quality segmentation with fewer architectural changes, but it was not considered or evaluated in the paper.

**Questions:**

1.	Could the authors elaborate on why they chose to couple the generation of 3D shapes and segmentation labels instead of generating shapes first and then applying segmentation?
	2.	How does the joint generation process improve upon using a standalone 3D segmentation model with an additional classification head?
	3.	Could the authors clarify how they address the potential mismatch in probability distributions between 3D points and segmentation labels?

---

### Note · Authors · 2024-11-14

I have read and agree with the venue's withdrawal policy on behalf of myself and my co-authors.